# Nb-induced lattice changes to enhance corrosion resistance of $Al_{0.5}Ti_3Zr_{0.5}Nb_xMo_{0.2}$ high-entropy alloys

Xuelian Yu, Qingjun Chen ✉, Xia Cui & Delai Ouyang

In this work, the effect of lattice structure on the corrosion behavior and passivation film properties of reinforced $Al_{0.5}Ti_3Zr_{0.5}Nb_xMo_{0.2}$ (x = 0.5,0.8,1) high-entropy alloys are investigated. A single-phase BCC $Al_{0.5}Ti_3Zr_{0.5}Nb_xMo_{0.2}$ (x = 0.5, 0.8, 1) high-entropy alloys, exhibiting good corrosion resistance, are synthesized using vacuum arc melting. Nb improves the corrosion resistance of high-entropy alloys in two main ways. On the one hand, the alloys show preferential corrosion at the {011} crystalline planes. Increasing Nb content reduced the {011} crystalline plane spacing, enhancing the corrosion resistance of $Al_{0.5}Ti_3Zr_{0.5}NbMo_{0.2}$. On the other hand, during the corrosion process, Nb, which has a large atomic radius and strong oxygenophilicity, interacts with each metal element, contributing to the uphill diffusion of Al/Ti and the downhill diffusion of O. The low-valent oxides form first continuously react with the inward-diffusing O to form high-valent oxides. This results in the formation of a layered passivation film with high breakdown potential and high stability. This work provides a basis for designing chemically robust alloys for extreme environments.

Traditional alloys, such as stainless steel, commonly employed in marine vessels and the nuclear industry, are prone to localized corrosion and premature failure. The corrosion resistance of alloys is closely related to the microstructure of the alloy and the composition and structure of the passivation film. High-entropy alloys, an emerging class of alloys, have garnered significant attention since their introduction in 2004[1,2]. Unlike traditional alloys, high-entropy alloys are more inclined to form single-phase disordered solid solutions, such as face-centered cubic (FCC)[3,4], body-centered cubic (BCC)[5,6], and densely-rowed hexagonal (HCP)[7,8] structures due to the higher mixing entropy, which ensures the homogeneity of the chemical elements in all parts of the alloy. This unique design endows high-entropy alloys with superior corrosion resistance, mechanical properties[9–12], and high-temperature oxidation resistance[13–16] compared to conventional alloys. In recent years, the corrosion behavior of various different systems of high-entropy alloys in different corrosive environments has been studied, including NaCl, HCl, $H_2SO_4$, and NaOH solutions. Among them, FeCoCrNi based high entropy alloy is the most widely studied system among the FCC high entropy alloy systems[17–22]. On the basis of this system, the addition of beneficial elements Mo, Nb, etc. can

improve the content of corrosion-resistant oxides in the passivation film. In particular, the addition of Mo element increases the proportion of $Cr_2O_3$ in the passivation film without destroying the microstructure of high-entropy alloys, thus enhancing the pitting resistance of these high-entropy alloys in aqueous chlorine-containing solutions[23]. The addition of Nb element decreases the $I_{corr}$(Self-corrosion current density) and increases the $E_{corr}$(self-corrosion potential) by refining the grain size and changing the composition of the passivation film, while the addition of Nb reduces the pitting nucleation and pitting expandability[24].

The concept of refractory high entropy alloys (RHEA) was proposed in 2010, which consists of refractory metal elements V, Cr, Nb, Mo, Zr, Hf, Ta, W, etc. MoNbTaW and MoNbTaVW refractory high entropy alloys developed by O.N. Senkov et al. exhibit excellent high temperature properties, and their high temperature mechanical properties are greatly exceeded by those of high-temperature alloys[25], so RHEA is expected to be used in the industries such as nuclear power and coal combustion. Compared with FeCoCrNi-based high-entropy alloys, the corrosion resistance of RHEA is relatively less studied. Some current studies have shown that the corrosion resistance of RHEA is

School of Materials Science and Engineering, Nanchang Hangkong University, Nanchang, China. ✉e-mail: qjchen@nchu.edu.cn

superior to that of conventional alloys (stainless steel, nickel-based alloys, etc.). Some published refractory high-entropy alloys such as $Al_{0.1}CrNbSi_{0.1}TaTiV$ and $VCrFeTa_{0.2}W_{0.2}$[26,27] have excellent pitting resistance in chloride environments. In general, the excellent corrosion resistance of refractory high-entropy alloys is attributed to the formation of physical phases and the tendency of refractory elements to form stable passivation films. Gao et al.[28] used vacuum arc melting to prepare CrNbTaTiV RHEAs consisting of BCC matrix, fine Laves phases with $Cr_2Nb$ structure at grain boundaries, and Laves phases at grain boundaries will be galvanic coupling corrosion with the main phase of BCC. Chen et al.[29] used laser melting technique to prepare $CrFeNbTiMo_x$ (x = 0.2, 0.4, 0.6, 0.8, 1) refractory high entropy alloys (RHEAs) coatings on 40Cr surface, which consisted of BCC, C14-Laves, C15-Laves, and ordered $B_2$ phases, and the corrosion resistance of the coatings was significantly improved with the increase of the Mo content, increase of the BCC phase, and the decrease of the Laves phase. Zhang et al.[30] used multi-target DC magnetron co-sputtering technique to prepare Nb-Ta-W-Hf high-entropy alloy film with amorphous structure, the corrosion current density of the film is about two orders of magnitude lower than that of 304 stainless steel, while the polarization resistance is much higher than that of 304 stainless steel. The surface roughness of $Ti_{1.5}Al_{0.3}ZrNb$ RHEA prepared using laser shock processing (LSP) increased, and significant grain refinement occurred on the surface from micron to nanometer grains, and this surface morphology significantly improved the corrosion resistance of RHEA in simulated body fluids[31]. Gao et al.[5] investigated the passivation behavior of $Mo_{15}Nb_{20}Ta_{10}Ti_{35}V_{20}$ high entropy alloy by using Mott-Schottky curve analysis and XPS, which showed that the passivation film of RHEA in 3.5% NaCl solution showed the double layer structure feature, which explains the phenomenon of secondary passivation. Wang et al.[32] investigated the passivation behavior of x-Zr-50Ti-(50-x) Nb (x = 40, 34, 26 and 20%) meso-entropic alloy in 0.5 M $H_2SO_4$ + 5 ppm NaF at 70 °C. The main components of the passivation film were $ZrO_2$, $TiO_2$ and $Nb_2O_5$, and the doping of Zr and Nb effectively hindered the formation of defects in the passivation film.

The above studies were conducted to enhance the corrosion resistance of the existing high-entropy alloys by changing the alloy composition, preparation process and adding corrosion-friendly elements, but the enhancement was limited, and the formation process of passivation film and its internal composition and structure were seldom involved, and the corrosion resistance of the materials was affected by the atomic composition of the substrate, the substrate organization, and the structure of the passivation film. Based on the above considerations, a highly corrosion-resistant $Al_{0.5}Ti_3Zr_{0.5}Nb_xMo_{0.2}$ light-weight refractory high-entropy alloy was designed by selecting three refractory corrosion-resistant metal elements, Zr, Nb, and Mo, and additionally introducing the lightweight elements, Al and Ti, and at the same time, increasing the content of the passivation-prone element, Ti. The effect of Nb on the corrosion and passivation behavior of the as-cast $Al_{0.5}Ti_3Zr_{0.5}Nb_xMo_{0.2}$ high-entropy alloy in 3.5% NaCl solution at room temperature was investigated. The microstructure of the high entropy alloy was characterized by X-ray diffractometer (XRD), scanning electron microscope (SEM) and transmission electron microscope (TEM). The corrosion behavior of the high-entropy alloy was investigated by various electrochemical tests, and the structure and composition of the passivation film were systematically characterized by using SEM, TEM, and X-ray photoelectron spectrometer (XPS), etc. The corrosion mechanism of $Al_{0.5}Ti_3Zr_{0.5}Nb_xMo_{0.2}$ high-entropy alloy in the environment of 3.5% NaCl solution was elaborated.

## Results
### Structural and microstructural characterization of HEA
The formation of high entropy alloys is related to thermodynamic parameters such as $\Delta S_{mix}$ (entropy of mixing), $\Delta H_{mix}$ (enthalpy of mixing) VEC (valence electron concentration), and δr (difference in

atomic radii)[33–35], which can be calculated by the following equation:

$$\Delta S_{mix} = -R\left[c_1 \ln c_1 + \cdots + c_n \ln c_n\right] = -R\sum_{i=1}^{n} c_i \ln c_i \quad (1)$$

$$\Delta H_{mix} = \sum_{i=1, i\neq j}^{n} \Omega_{ij} c_i c_j = 4\sum_{i=1, i\neq j}^{n} \Delta H_{ij}^{mix} c_i c_j \quad (2)$$

$$\delta_r = \sqrt{\sum_{i=1}^{n} c_i \left(1 - \frac{r_i}{\bar{r}}\right)^2}, \quad \bar{r} = \sum_{i=1}^{n} c_i r_i \quad (3)$$

$$VEC = \sum_{i=1}^{n} c_i (VEC)_i \quad (4)$$

$$\Omega = \frac{T_m \Delta S_{mix}}{|\Delta H_{mix}|} \quad (5)$$

$$\Delta \chi = \sqrt{\sum_{i=1}^{n} c_i (x_i - \bar{x})^2}, \quad \bar{x} = \sum_{i=1}^{n} c_i x_i \quad (6)$$

where R (8.314 J/K/mol) is the gas constant, $c_i$ is the molar ratio of the ith element, $H_{ij}$ is the enthalpy of mixing of the ith element with the jth element, $\bar{r}$ is the average atomic radius, and $\Omega$ is the parameter used to predict the formation of solid solution.

The thermodynamic parameters calculated from the above equations are listed in Table S1. The atomic radii and electronegativity of the five metallic elements are listed in Table S2. The lattice changes of the high-entropy alloy with the change of Nb are characterized by $\delta_r$. From Table S1, the atomic radius differences of the high-entropy alloy $Al_{0.5}Ti_3Zr_{0.5}Nb_xMo_{0.2}$ are 3.06%, 3.11%, and 3.2%, respectively.

Figure 1 illustrates the X-ray diffraction (XRD) patterns of the high-entropy alloy $Al_{0.5}Ti_3Zr_{0.5}Nb_xMo_{0.2}$, which reveals that all three samples exhibit diffraction peaks at approximately 2θ angles of 39°and 56°. Furthermore, diffraction peaks were observed at 2θ angles of 39°, 56°, and 70°, corresponding to the (110), (200), and (211) crystal planes, respectively. These observations indicate that the prepared high-entropy alloy $Al_{0.5}Ti_3Zr_{0.5}Nb_xMo_{0.2}$ exhibits a single-phase BCC structure. Rietveld refinement was applied to the samples, and the resulting parameters are presented in Fig. 1. The data therein demonstrate that the space groups of the high-entropy alloys are all Im-3m, which can also be identified as a BCC structure. Furthermore, the lattice constant of the high-entropy alloys exhibit a slight increase with rising Nb content, which can be attributed to the fact that doping more Nb atoms with larger atomic radii will result in a larger lattice[36].

From the SEM-EDS mapping of the high-entropy alloy (Fig. 2), it can be seen that the chemical distribution of the five elements is uniform and does not aggregate along the grain boundaries. Therefore, the elemental chemical distribution does not play an influential role in the corrosion process.

Figure 3 shows the TEM images of the high entropy alloy $Al_{0.5}Ti_3Zr_{0.5}Nb_xMo_{0.2}$. According to the high-resolution images and Fast Fourier Transform (FFT) of the high-entropy alloy $Al_{0.5}Ti_3Zr_{0.5}NbMo_{0.2}$ in Fig. 3b, c, the crystal plane spacing of (002) is 1.652 Å and (110) is 2.282 Å. The corresponding selected area electron diffraction (SAED) image (Fig. 3a) identifies the crystallographic band axis as [011] and also shows the presence of BCC solid solution, which is consistent with the XRD results. Similarly, Fig. 3d–f shows that the (110) crystal plane spacing of the high entropy alloy $Al_{0.5}Ti_3Zr_{0.5}Nb_{0.8}Mo_{0.2}$ is 2.289 Å and the (200) crystal plane spacing is 1. 632 Å. The (110) crystal plane spacing of the high entropy alloy $Al_{0.5}Ti_3Zr_{0.5}Nb_{0.5}Mo_{0.2}$ is 2.295 Å and the (200) crystal plane spacing is 1.614 Å.

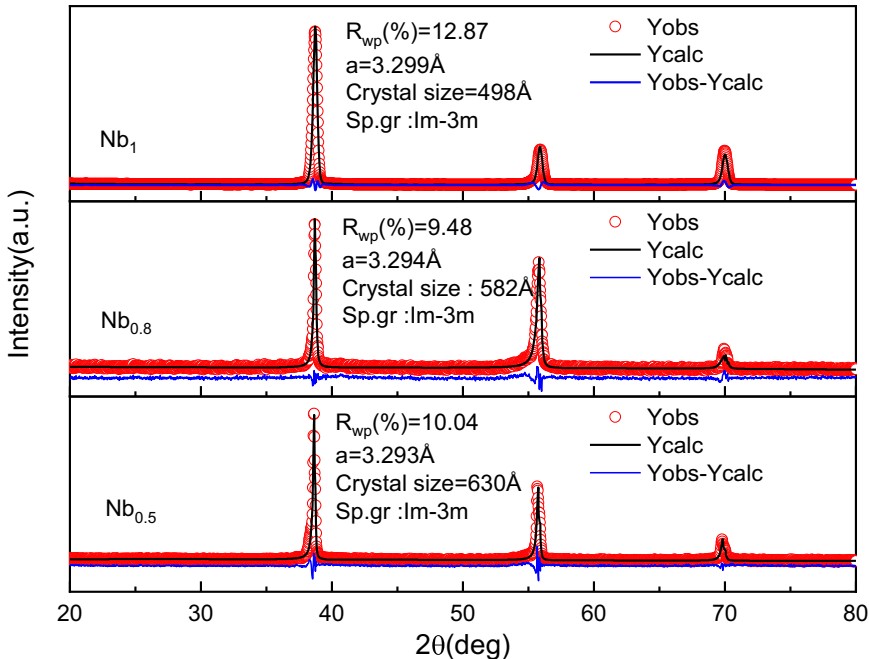

**Fig. 1 | XRD Rietveld refinement pattern of high entropy alloy Al$_{0.5}$Ti$_3$Zr$_{0.5}$Nb$_x$Mo$_{0.2}$.** R$_{wp}$, Goodness of fit (GF), a, lattice constant, Sp.gr, space group. Source data are provided as a Source Data file.

## Corrosion behavior

Figure 4a illustrates the room temperature kinetic potential polarization curves of the high entropy alloys in 3.5% NaCl solution. An abrupt increase in current density occurs upon entering the dissolution zone from the primary passivation zone. This rise stabilizes shortly thereafter. This indicates that Al$_{0.5}$Ti$_3$Zr$_{0.5}$Nb$_x$Mo$_{0.2}$ high-entropy alloys have a strong re-passivation capability. The electrochemical parameters were calculated using the extrapolation method and are presented in Table S3. It can be observed that the three compositions of high-entropy alloys exhibit higher self-corrosion potentials and lower self-corrosion current densities. Furthermore, the extensive, stable passivation zone and the markedly elevated breakdown potential of 8.68 V$_{SCE}$ suggest that the high-entropy alloy exhibits excellent corrosion resistance. It displays superior corrosion resistance compared to nickel, zirconium, iron-based alloys[37–39], and other high-entropy alloys in a 3.5% NaCl solution at room temperature. Comparison of E$_{pit}$-I$_{corr}$ plots of corrosion of Al$_{0.5}$Ti$_3$Zr$_{0.5}$NbMo$_{0.2}$ high-entropy alloy with titanium alloys[40–42], nickel-based alloys[43,44], and high-entropy alloys[6,29,45–51] such as FeCoCrNiMn[52,53] system and NbMoZrTiAlV$_x$ system[47,54] corroded under 3.5% NaCl solution are listed in Fig. 4b.

## Structure and composition of passivation films

In order to study the composition and structure of the passivation films generated by primary and secondary passivation, the Al$_{0.5}$Ti$_3$Zr$_{0.5}$Nb$_x$Mo$_{0.2}$ high-entropy alloy was polarized at constant potential for 40 min. The cross-section of the passivation film on the surface of the specimen was further observed using SEM following constant potential polarization for 40 min at 4 V$_{SCE}$ in a 3.5 wt.% NaCl solution. As can be observed in Fig. 4c, d, a clear passivation film layer is present. Figure 4d demonstrates a cross-section of the sample magnified by 2000 times, which shows that the surface of the passivation film has built-up products. This indicates that the corrosion products generated by the secondary passivation will block the ion channels and inhibit the corrosion reaction. From Fig. 4e, it can be observed that with increasing depth, the concentration of five elements, Al, Ti, Zr, Nb and Mo, initially increased and then reached a plateau. Subsequently, the concentration of the O element decreased, indicating the presence of a passivation film with a thickness of ~1 μm[55].

As illustrated in Fig. 4f, the surface subjected to the dynamic potential polarization test exhibits a distribution of pits of varying sizes, which are enveloped by solid deposits. This phenomenon is postulated to be the consequence of the breakdown of the passivation film upon reaching the breakdown potential, resulting in pitting corrosion on the surface of the alloy. This process ultimately leads to the dissolution of the substrate due to the loss of the passivation film. The surface devoid of pits is relatively flat, indicating the formation of a continuous passivation film. The locations of the pits are randomly distributed, and the deposits within the pits are not entirely dislodged. The histomorphology of the pits is illustrated in Fig. 4g, which reveals the presence of discernible grain contours and pronounced cracks at the grain boundaries, as depicted in Fig. 4f. The interior of the grains is coated with an oxide layer, which serves to safeguard them from further corrosion. This observation suggests that the initial local corrosion has progressed to active corrosion, with the grain boundaries exhibiting preferential corrosion.

From the Fig. 5, it can be observed that the elements Nb, Ti, and Zr are present in the highest valence form on the surface of the passivated film, specifically Nb$^{5+}$ [24,56], Ti$^{4+}$ [41,57], and Zr$^{4+}$ [58], while the element Al is present in a minor quantity in the form of Al$^0$, in addition to the highest valence state Al$^{3+}$ [51,59]. The element Mo is predominantly present in the forms of Mo$^{6+}$ and Mo$^{4+}$ [23]. Figure 5h depicts the high-resolution XPS spectra of the O 1 s. From the surface O 1 s spectra, it can be observed that the O element on the sample surface has three splitting peaks in the 1 s orbitals, which include a significant amount of O$^{2-}$ as well as a minor amount of OH$^-$ and bound water[6,60,61]. This suggests that the surface of the passivation film is primarily composed of the highest valence oxides of each element, with a minor presence of hydroxides. Additionally, the adsorption of H$_2$O on the passivation film is highly facile and plays a pivotal role in the formation of amorphous or nanocrystalline titanium within the passivation film[41]. At an argon ion sputtering depth of 2.5 nm, the elemental composition of oxygen is observed to consist solely of lattice oxygen ions, with a minor presence of bound water. The Ti element in the passivation film has added Ti$^{3+}$ and a small amount of Ti$^{2+}$, while the Nb element has added Nb$^{4+}$ ions on the basis of the presence of Nb$^{5+}$ ions. The Zr and Al elements remain basically unchanged in terms of the types and contents of each

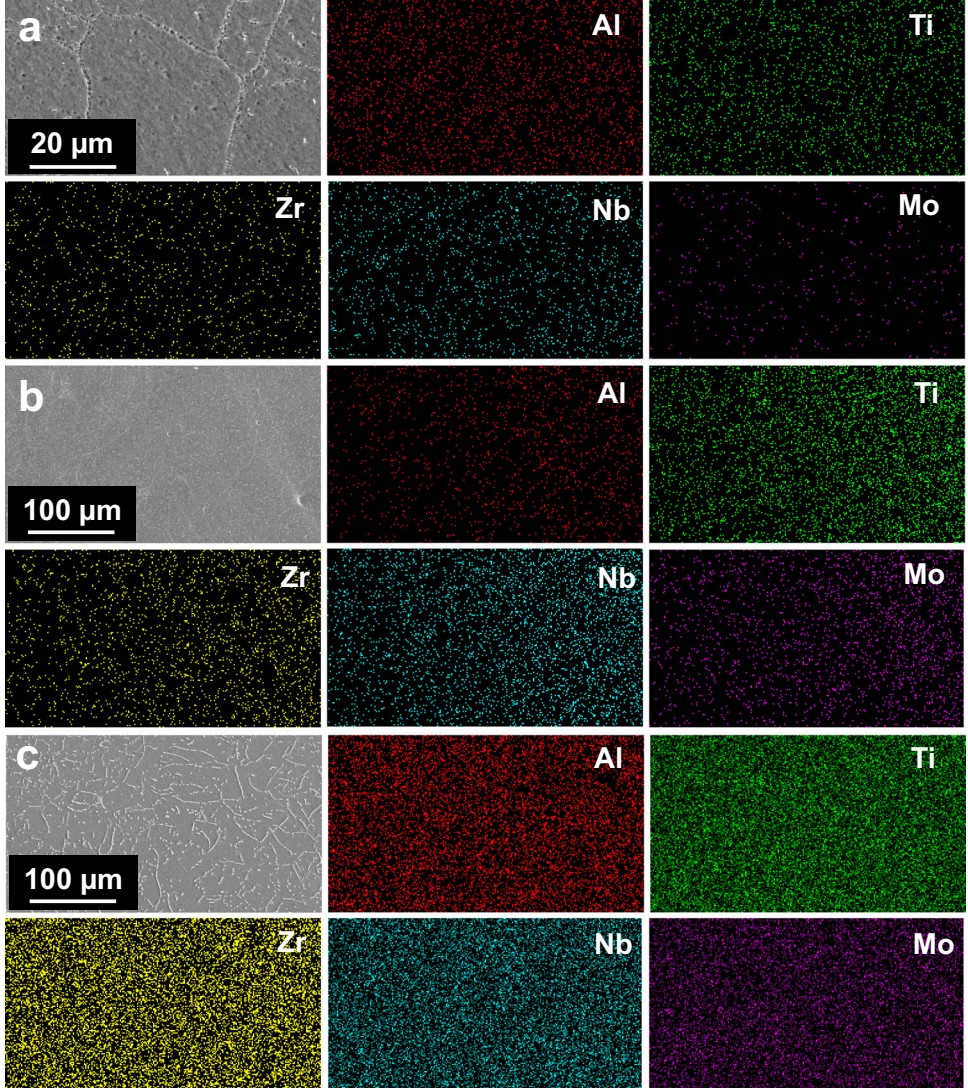

**Fig. 2 | SEM-EDS mapping of high-entropy alloys showing a uniform distribution of elements with no aggregation at grain boundaries. a** $Al_{0.5}Ti_3Zr_{0.5}NbMo_{0.2}$,
**b** $Al_{0.5}Ti_3Zr_{0.5}Nb_{0.8}Mo_{0.2}$, **c** $Al_{0.5}Ti_3Zr_{0.5}Nb_{0.5}Mo_{0.2}$. Source data are provided as a Source Data file.

valence state. Conversely, the $Mo^{6+}$ decreases while the $Mo^{4+}$ increases. This suggests that the passivation film is primarily composed of the highest valence oxides of each metal element, with a minor contribution from low valence oxides, across the surface and inner depth of 2.5 nm. Upon reaching an argon ion sputtering depth of 10 nm, the passivation film is observed to incorporate $Nb^{2+}$ and $Mo^0$, while $Mo^{6+}$ is no longer present and the remaining elements exhibit no valence alterations. At an argon ion sputtering depth of 20 nm, $Zr^0$ is observed in the passivation film, accompanied by an increase in $Nb^{2+}$, $Ti^{2+}$, and $Al^0$ content, and a concomitant decrease in the concentration of high-valence cations, including $Nb^{5+}$, $Ti^{4+}$, and $Ti^{3+}$.

Semi-quantitative analyses of the valence ions were conducted based on the results of the fine spectral analysis of each element, and the changes in the content are presented in Fig. 5b and Table S4. The elemental contents of Al, Ti, Zr and Mo on the surface of the passivated film were 12.96%, 61.82%, 9% and 7.21%, respectively. The elemental percentage of Nb was 9.12%, which was reduced by half compared to the atomic percentage content of each element in the matrix of the $Al_{0.5}Ti_3Zr_{0.5}NbMo_{0.2}$ high-entropy alloy. The other four elemental percentages were all different, with the percentage content of the other four elements exhibiting varying degrees of elevation. The same conclusion was reached by calculating the ion content of the

passivation film at argon ion sputtering depths of 2.5 nm, 10 nm, and 20 nm. This indicated that the oxides of each metal element competed for growth in the corrosive environment of this condition. The passivation film at all depths was predominantly composed of Ti's oxides, with Nb's oxides growing more slowly than those of other elements.

As illustrated in Fig. 5, the outer layer of the passivation film is primarily constituted by high-valent oxides of each metal element. The concentration of high-valent oxides gradually diminishes from the outer layer to the inner layer, while the low-valent oxides exhibit a gradual decrease. Notably, the oxide content of the Al and Zr elements shows minimal variation, suggesting that the passivation film is predominantly composed of the $Al_{0.5}Ti_3Zr_{0.5}NbMo_{0.2}$ high-entropy alloy exhibits a layered structure when immersed in a 3.5% NaCl solution[5].

In accordance with the elemental analysis of the primary passivation film illustrated in Figs. 5, 6 presents the XPS analysis of the secondary passivation film. The high-resolution XPS patterns of the Nb, Ti, Zr, Al and Mo elements are presented in Fig. 6c–g. From the figure, it can be observed that high-valent ions, including $Ti^{4+}$, $Nb^{5+}$, $Zr^{4+}$, $Mo^{6+}$ and $Al^{3+}$ are present on the surface of the secondary passivated film, which also contains a minor quantity of 0-valent aluminium. The O 1 s spectra indicate that the passivated film is predominantly composed of metal oxides, with a minor contribution from hydroxides

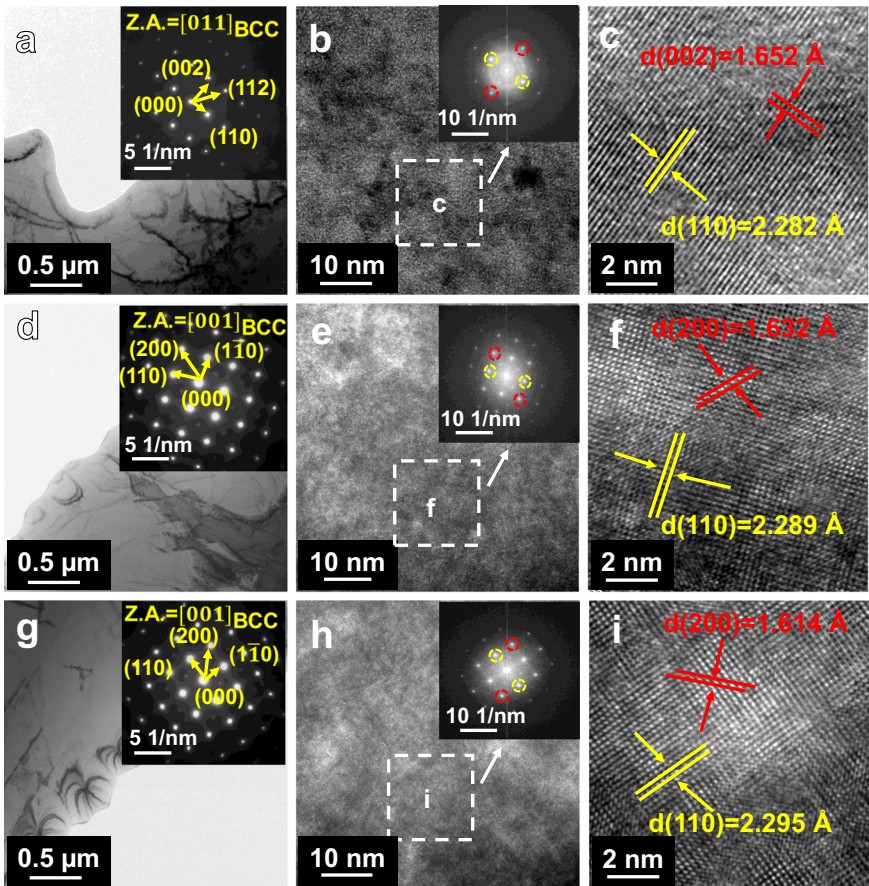

**Fig. 3 | TEM patterns of the as-cast high-entropy alloy Al$_{0.5}$Ti$_3$Zr$_{0.5}$Nb$_x$Mo$_{0.2}$.** **a**–**i** are the bright-field images, high-resolution maps, and high-resolution magnified images of the high-entropy alloys Al$_{0.5}$Ti$_3$Zr$_{0.5}$NbMo$_{0.2}$, Al$_{0.5}$Ti$_3$Zr$_{0.5}$Nb$_{0.8}$Mo$_{0.2}$, Al$_{0.5}$Ti$_3$Zr$_{0.5}$Nb$_{0.5}$Mo$_{0.2}$, respectively. Red markers in (**c**, **f**, **i**) representing the (002) crystal plane and yellow markers representing the (110) crystal plane. Source data are provided as a Source Data file.

and bound water. This is in accordance with the observed composition of the primary passivated film. At an argon ion sputtering depth of 2.5 nm, the OH$^-$ species is no longer present, a small quantity of Ti$^{3+}$ is observed, and the Mo$^{4+}$ concentration rises, while the remaining ion types and quantities remain largely unaltered. This suggests that the passivation film is primarily composed of high-valent oxides of various elements. Upon argon ion sputtering to a depth of 10 nm, the presence of Nb$^{4+}$ and Ti$^{2+}$ was observed in the passivation film, while the number and types of other ions remained largely unaltered. Upon argon ion sputtering to a depth of 20 nm, the passivation film exhibited the appearance of Ti$^{2+}$, accompanied by an increase in the content of Nb$^{4+}$ and Mo$^{4+}$, a decrease in the content of Ti$^{4+}$, Nb$^{5+}$ and Mo$^{6+}$, and no discernible change in the content of Zr$^{4+}$ and Al$^{3+}$.

Semi-quantitative analyses of the elements in the passivated films were performed based on the results of fine spectroscopy, and the content of each ion was amounts were recorded in Fig. 6b with Table S4. The elemental contents of Al, Ti, Zr and Mo on the surface of the secondary passivated film are 11.29%, 68.82%, 8.41%, 2.76% and 8.74% of Nb, respectively. Compared with the atomic percentage content of each element in Al$_{0.5}$Ti$_3$Zr$_{0.5}$NbMo$_{0.2}$ high-entropy alloy substrate, the percentage of Nb is reduced by half, that of Mo by 1.34%, and the atomic percentage content of the other three elements is reduced by 1.34%. 1.34%, and the other three elements have different degrees of increase in percentage content. Compared to the primary passivated film surface, the elemental atomic content of Ti increased by 7%, Mo decreased by 4.45%, and the other elemental percentages remained basically unchanged. This indicates that the oxide growth rate of Ti is faster after entering the secondary passivation zone, and the oxide growth rate of Mo becomes slower.

As illustrated in Fig. 6, the surface of the secondary passivation film is identical to that of the primary passivation film. The latter is constituted by the highest valence oxides of each element, a minor quantity of hydroxide, and a minor quantity of bound water. The primary passivation film exhibits a low content of ions in the high valence state of each element, such as Nb$^{5+}$ and Ti$^{4+}$, as evidenced by the argon ion sputtering depth of 20 nm. The low valence state ions are present in high concentrations, including Nb$^{2+}$ and Ti$^{2+}$. Additionally, the elements Zr and Mo are observed in the 0-valence state. In contrast, the secondary passivation film is still characterized by a prevalence of high valence ions, which suggests that the secondary passivation film is of greater thickness than the primary passivation film. The secondary passivation film is primarily constituted by high-valent oxides within the tested depth range.

In order to study the structure of the passivation film, the secondary passivation film was analyzed using TEM. Figure 7a shows the bright field image observed in the thin zone after ion thinning, and a large number of nanocrystals with a size of about 10 nm can be observed. Comparison of the bright field image of the high entropy alloy matrix in Fig. 3a shows that the nanocrystals observed at this magnification belong to the structure of the passivation film generated by constant potential polarization. The analytical results of the selected area electron diffraction are displayed in Fig. 7b, where the diffraction rings correspond to the individual crystal surfaces of TiO$_2$ (rutile type), Nb$_2$O$_5$ and Al$_2$O$_3$, respectively[62]. In order to further analyze the passivation films, high-resolution transmission analysis was performed on the passivation films. From Fig. 7c, it can be seen that in addition to the presence of crystals, there is a small amount of amorphous. The crystal plane spacing $d = 0.248$ nm in Fig. 7d corresponds to the (101) crystal plane of rutile TiO$_2$, which is in agreement with the

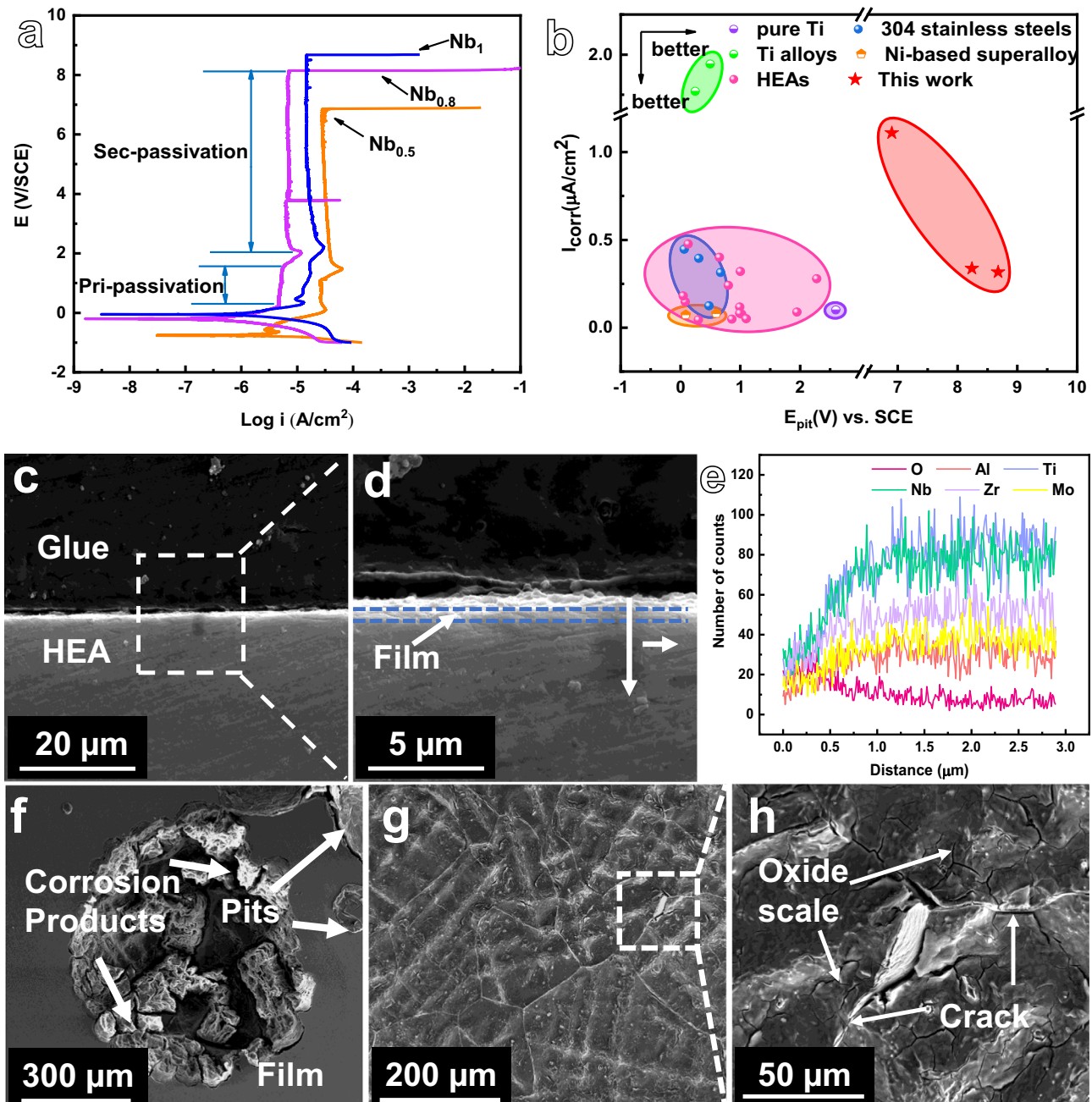

**Fig. 4 | Electrochemical testing, corrosion surface and cross-section morphology of $Al_{0.5}Ti_3Zr_{0.5}Nb_xMo_{0.2}$ high entropy alloy in 3.5% NaCl solution. a** Kinetic potential polarization curves. **b** Comparison of corrosion kinetic parameters $I_{corr}$ vs $E_{pit}$ for some reported high entropy alloys and some conventional alloys in 3.5% NaCl environment. **c** cross-section of the sample. **d** zoomed in on the labeled area of (**c**). **e** line scan EDS of the labeled area of (**d**). **f** surface SEM 300x. **g** corrosion pit interior SEM 500x. **h** corrosion pit interior SEM 2000x. Source data are provided as a Source Data file.

results of the selected electron diffraction analysis in Fig. 7b. Figure 7e with crystal plane spacing $d = 0.2054$ nm is identified as the (210) crystal plane of $TiO_2$, while the crystal plane spacing $d = 0.202$ nm is determined to correspond to the (21$\bar{1}$) crystal plane of $Al_2O_3$. The high-resolution transmission analysis demonstrates that the passivation film at the observed site comprises the highest valence oxides of each element in the matrix. In conjunction with the XPS results, it can be inferred that the observed site represents the outermost structure of the passivation film. Given the affinity of the element Al and Ti for oxygen and the stability of $Al_2O_3$ and $TiO_2$, $Al_2O_3$ and $TiO_2$ is preferentially generated on the surface of the passivation film.

Figure 8a depicts the cross-sectional (TEM) image of the high-entropy alloy following corrosion. It can be observed that the thickness of

the passivation film is approximately 11 nm, and the cross-sectional structure is amorphous. In conjunction with the XPS and cross-sectional HAADF and EDS analyses presented in Fig. 8b, it becomes evident that the oxide nanoparticles of Ti are dispersed within the passivation film, with a greater degree of uniformity observed in the regions closer to the surface. Figure 8 illustrates that the Me/F interface becomes undulating following corrosion, with undulations ranging from 2 to 5 nm. Additionally, sporadic deeper pits are observed. Furthermore, the high-resolution images and FFT analysis indicate that the undulating interface walls are parallel to the {002} and {112} crystal planes (Fig. 8c, d), suggesting that the corrosion process is strongly influenced by the anisotropy of the crystal orientation. From the selected area electron diffraction (SAED) in Fig. 8a, the crystallographic band axis is [011], indicating that the

corrosion rate along the {011} crystallographic plane is faster than that along the {002} and {112} crystallographic planes. Some intact {002} and {112} crystallographic planes are observed at the Me/F interfacial bulge, suggesting that the rapid dissolution of the {011} crystallographic plane determines the corrosion tendency and rate of the alloy.

## Discussion

### Nb-induced reduction in {011} crystal plane spacing reduces the corrosion rate

The dissolution of the matrix along the {011} crystal plane is the determining factor in the surface corrosion of the high-entropy alloy

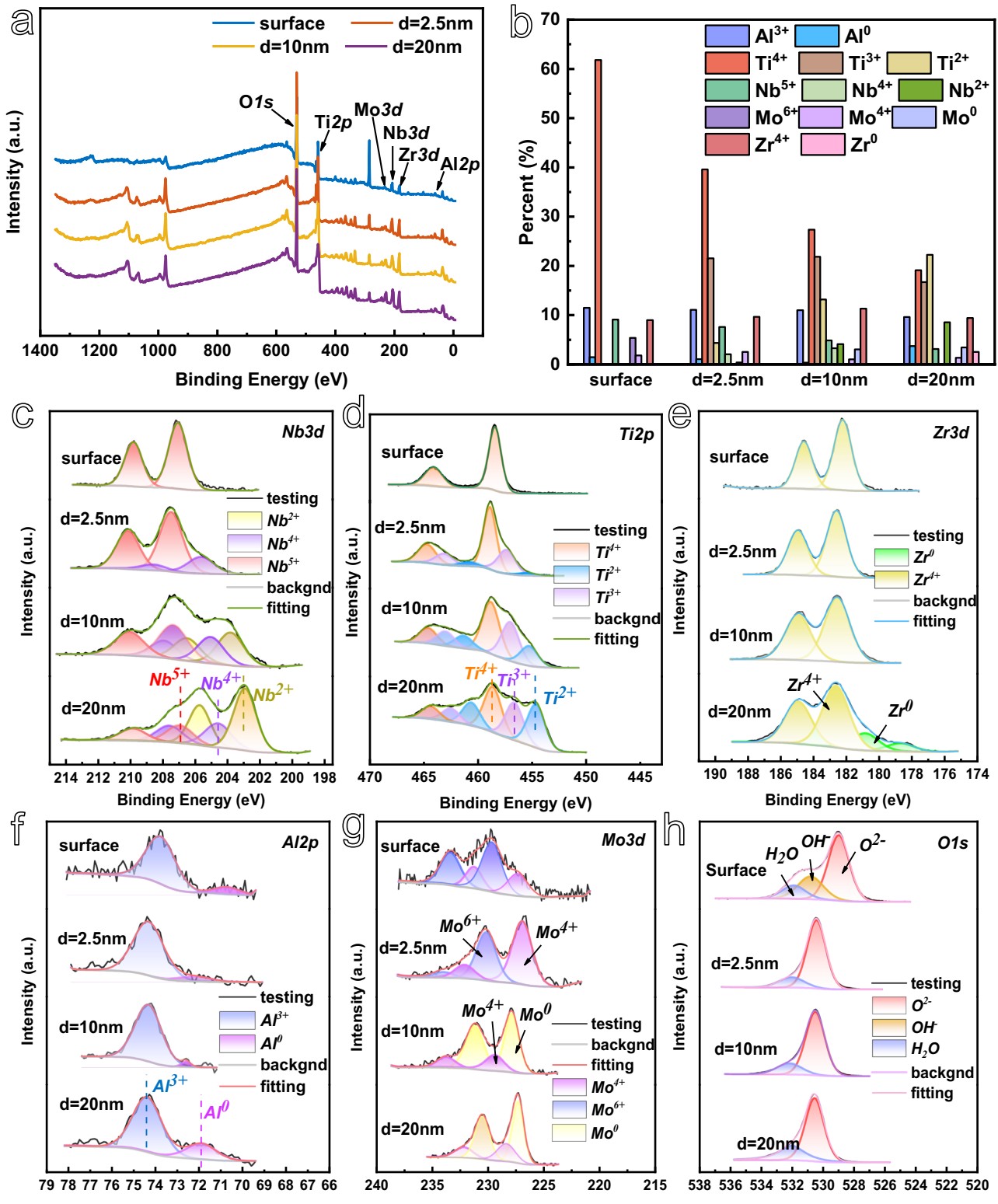

**Fig. 5 | XPS analysis of different sputtering depths of the passivation film on the surface of Al$_{0.5}$Ti$_3$Zr$_{0.5}$NbMo$_{0.2}$ high-entropy alloy in 3.5 wt.% NaCl solution with 1 V$_{SCE}$ constant potential polarization for 40 min of specimen at room** temperature. **a** XPS full spectrum. **b** Different valence ions content in the passivation film. High resolution XPS spectra of **c** Nb. **d** Ti. **e** Zr. **f** Al. **g** Mo. **h** O. Source data are provided as a Source Data file.

$Al_{0.5}Ti_3Zr_{0.5}Nb_xMo_{0.2}$. Consequently, the {011} crystal plane spacing is 2.295 Å, 2.2289 Å, and 2.282 Å, respectively. Interactions between elements cause changes in the crystal lattice. The atomic radii of five elements, Al, Ti, Zr, Nb and Mo, are in the order of r(Al) < r(Ti) < r(Mo) < r(Nb) < r(Zr), and the thermodynamic parameters

of the five elements solidly dissolved in a certain ratio, such as $\Delta S_{mix}$ (entropy of mixing), $\Delta H_{mix}$ (enthalpy of mixing), VEC (valence electron concentration), and $\delta_r$ (difference in atomic radius), were calculated and supplemented in Table S1. where the atomic radius difference $\delta_r$ is 3.2%, 3.11%, and 3.06% when x = 0.5, 0.8, 1, respectively. This indicates that

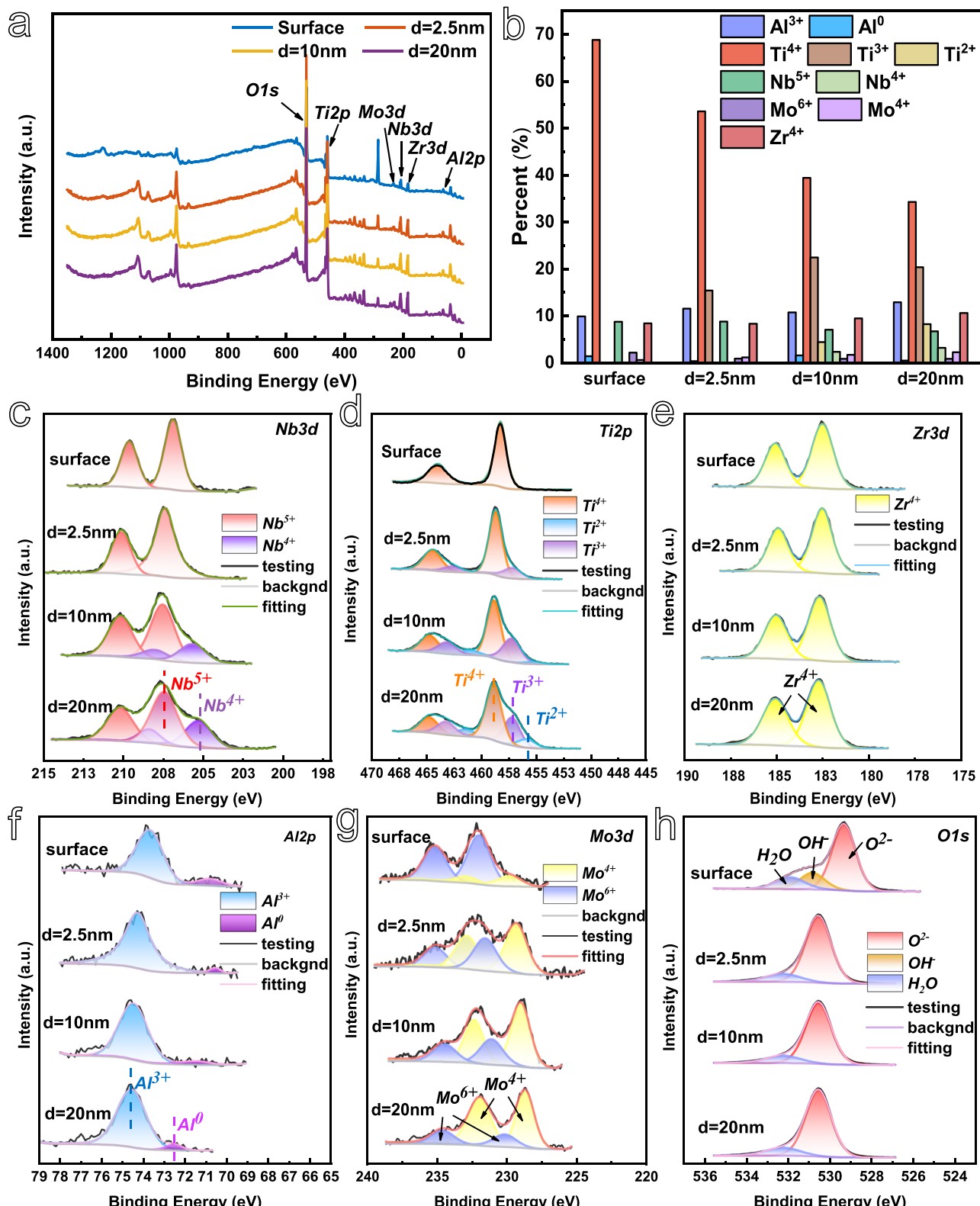

**Fig. 6 | XPS analysis of different sputtering depths of the passivation film on the surface of $Al_{0.5}Ti_3Zr_{0.5}NbMo_{0.2}$ high-entropy alloy in 3.5 wt.% NaCl solution with 4 $V_{SCE}$ constant potential polarization for 40 min of specimen at room** temperature. **a** XPS full spectrum. **b** Different valence ions content in the passivation film. High resolution XPS spectra of **c** Nb. **d** Ti. **e** Zr. **f** Al. **g** Mo. **h** O. Source data are provided as a Source Data file.

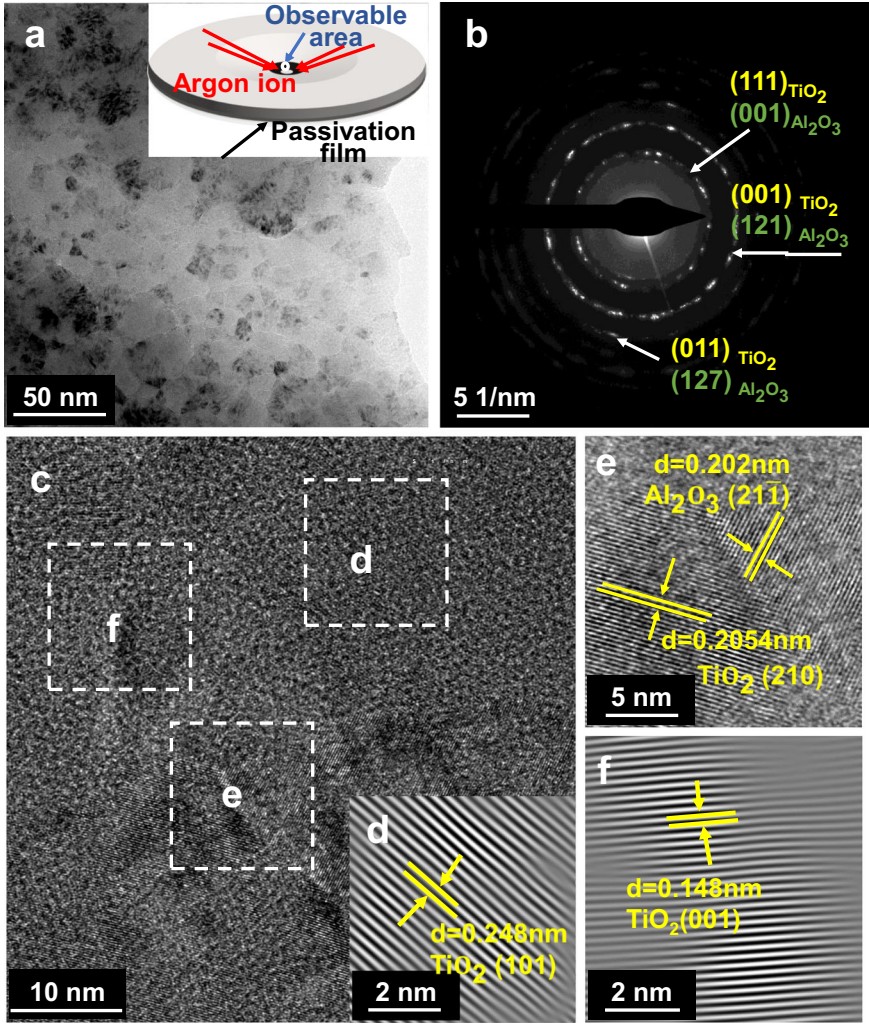

**Fig. 7 | Surface TEM images of passivation film of high entropy alloy $Al_{0.5}Ti_3Zr_{0.5}NbMo_{0.2}$ after constant potential polarization at 4 $V_{SCE}$ for 40 min. a** surface bright-field image. Inset showing a schematic diagram of one-sided ion thinning to obtain the most superficial passivation film. **b** SAED of the most surface corrosion products. **c** High resolution image of the surface passivation film. **d** surface high-resolution and inversion Fast Fourier transformation (IFFT) of $TiO_2$ (101). **e** surface high-resolution of $TiO_2$ (210) and $Al2O3$ (21$\bar{1}$). **f** surface high-resolution and inversion Fast Fourier transformation (IFFT) of $TiO_2$ (001). Source data are provided as a Source Data file.

increasing the molar content of Nb decreases the atomic radius difference of the high-entropy alloy, reducing the extent of lattice distortion and therefore changing the crystal plane spacing of (110). when x = 0. 5, 0.8, 1 (Fig. 3), and with the increase of the Nb content, the {011} crystal plane spacing is observed to decrease gradually, resulting in a corresponding increase in the corrosion potential of the three compositions of high-entropy alloys (Fig. 4a). This is due to the fact that the larger crystal plane spacing provides a greater number of channels for the corrosive fluid, which can more easily penetrate the material surface and act upon it with greater efficacy. This results in a reduction in the corrosion tendency and rate of the material. This can also be illustrated by the observation that the high-entropy alloy at x = 0.5 has the largest corrosion current density of $1.11 \times 10^{-6}$ A/cm$^2$ and the lowest corrosion potential of −0.5787 $V_{SCE}$. Furthermore, the XRD Rietveld refinement results indicate that the grain size is 630 Å, 582 Å, and 498 Å when x = 0.5, 0.8, and 1, respectively. With the increase of Nb content, the grain size decreases and the lattice constants increase gradually. The density of the high-entropy alloy $Al_{0.5}Ti_3Zr_{0.5}Nb_xMo_{0.2}$ was determined through the drainage method, yielding densities of 5.39 g/cm$^3$, 5.27 g/cm$^3$, and 5.18 g/cm$^3$ at x = 0.5, 0.8, and 1, respectively. This suggests that as the lattice constant increases, the spacing between neighboring cells expands, the density of the crystals declines, and the alloy's corrosion resistance is enhanced[63,64].

## Thermodynamic mechanisms of competitive growth of passivation films

The kinetic potential polarization demonstrates that the single-phase BCC $Al_{0.5}Ti_3Zr_{0.5}Nb_xMo_{0.2}$ high-entropy alloy exhibits excellent corrosion resistance. From a thermodynamic perspective, the driving force of oxides generated by metal elements can be expressed by $\Delta_r G_m^{\ominus}$ for the reaction of metal with $H_2O$ and the formation of suboxides in the passivation. The formation of the film is attributed to the insufficient supply of oxidizing agents in the inner layer of the passivation film, and the inadequate reaction of the metal elements. Therefore, the equations should be balanced with $H_2O$. In the case of the titanium element, for example, the reaction formula can be expressed as follows:

$$Ti + H_2O = TiO + 2H^+ + 2e^- \quad \Delta_r G_m^{\ominus} = -276.198 kJ/mol \quad (7)$$

$$2/3Ti + H_2O = 1/3Ti_2O_3 + 2H^+ + 2e^- \quad \Delta_r G_m^{\ominus} = -240.913 kJ/mol \quad (8)$$

$$Ti + 2H_2O = TiO_2 + 4H^+ + 4e^- \quad \Delta_r G_m^{\ominus} = -414.443 kJ/mol \quad (9)$$

The standard Gibbs free energies $\Delta_f G_m^{\ominus}$ for the generation of TiO, $Ti_2O_3$ and $TiO_2$ in the passivation film have been calculated to be

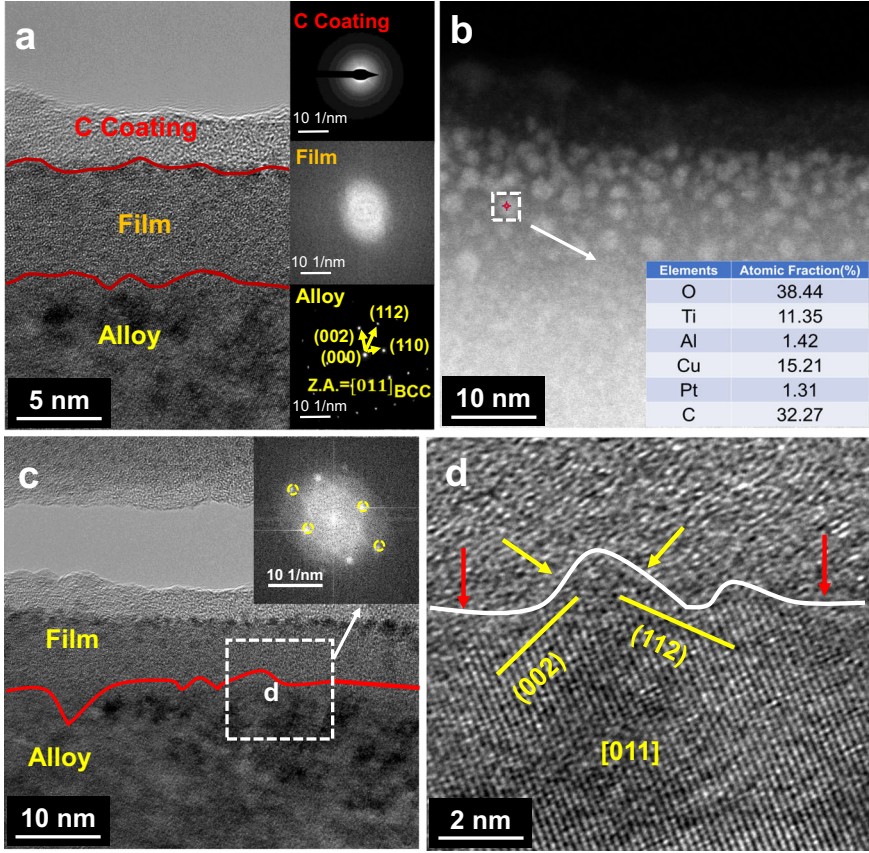

**Fig. 8 | Cross-sectional TEM image of the Al$_{0.5}$Ti$_3$Zr$_{0.5}$NbMo$_{0.2}$ high-entropy alloy after constant potential polarization at 4V$_{SCE}$ for 40 min. a** High-resolution images and SAED of C protective layer, passivation film and high-entropy alloy Al$_{0.5}$Ti$_3$Zr$_{0.5}$NbMo$_{0.2}$. **b** STEM-HAADF images and EDS of Cross-sectional oxides. **c** Undulating interface and FFT of the metal and the passivation film.

**d** Enlarged image of the interface. Yellow arrow representing the direction of corrosion on the raised portion of the (002) and (112) crystal plane. Red arrows representing the direction of corrosion along the (011) grain family, where the corrosion rate is faster. Source data are provided as a Source Data file.

−276.198 kJ/mol, −240.913 kJ/mol and −414.443 kJ/mol, respectively. It can be observed that the Ti element undergoes oxidation from its metallic state to TiO$_2$ in solution, with a more negative $\Delta_r G_m^{\Theta}$, indicating a greater propensity for TiO$_2$ generation in the outer layer of the passivation film. This conclusion is corroborated by the XPS results of the outermost layers of both the primary and secondary passivation films. Additionally, the highly reducing nature of TiO$_2$ facilitates its reaction with the bound water oxidant between the layers (Figs. 5h, 6h), leading to the formation of Ti$_2$O$_3$. To illustrate, the passivation film formation process for high entropy alloys in 3.5% NaCl solution can be expressed by the following equation:

$$Ti + H_2O = TiO + 2H^+ + 2e^- \quad \Delta_r G_m^{\Theta} = -276.198 kJ/mol \quad (10)$$

$$TiO + 1/2H_2O = 1/2Ti_2O_3 + H^+ + e^- \quad \Delta_r G_m^{\Theta} = -219.95 kJ/mol \quad (11)$$

$$TiO + H_2O = TiO_2 + 2H^+ + 2e^- \quad \Delta_r G_m^{\Theta} = -138.24 kJ/mol \quad (12)$$

$$1/2Ti_2O_3 + 1/2H_2O = TiO_2 + H^+ + e^- \quad \Delta_r G_m^{\Theta} = -53.073 kJ/mol \quad (13)$$

The reaction equation for the whole process is as follows:

$$Ti + 2H_2O = TiO_2 + 4H^+ + 4e^- \quad (14)$$

The reaction process of other elements is also similar, and the Gibbs free energies $\Delta_r G_m^{\Theta}$, which indicate the formation of all levels of

oxides of metal elements with H$_2$O, are listed in Table S5. From the table, it can be seen that the stable oxides Al$_2$O$_3$ and ZrO$_2$ with $\Delta_r G_m^{\Theta}$ of −870.7648 kJ/mol and −568.4432 kJ/mol, respectively, are directly reactive due to their more negative Gibbs free energies and their $\Delta_r G_m^{\Theta}$ are significantly lower than their respective suboxides[61]. They generate stable highest valence oxides without suboxides (Fig. 5e, f), Fig. 6e, f). Since Ti element accounts for 57.7% of the high-entropy alloy matrix, it dominates the competitive multi-element growth process. The MoO$_3$ has a $\Delta_r G_m^{\Theta}$ of 43.5352 kJ/mol, a more positive Gibbs free energy than other oxides and suboxides, and does not dominate in multi-elemental competition for growth. Consequently, passivation films contain a lower content of Mo oxides than matrix Mo.

**Nb induces atomic diffusion to form a layered passivation film**

The Nb concentration in the passivation film is 50% of the Nb concentration in the substrate, and the Ti/Al concentration in the passivation film is greater than the Ti/Al concentration in the substrate(Table S4), which is attributed to the fact that Nb has a larger atomic radius, resulting in a larger lattice, and under the action of the stress, the Ti/Al atoms of smaller atomic radii diffuse outward and combine with the inwardly diffusing O to form the passivation film, and after reaching a certain concentration, Ti/Al atoms diffuse uphill and O continues to diffuse downhill, resulting in the formation of a layered structure of the passivation film. According to the above SEM-EDS experimental results, it can be seen that the chemical distribution of the elements at each place of the high-entropy alloy is uniform, so at the initial moment of corrosion, the molar content of Ti on the surface and in the interior of the high-entropy alloy Al$_{0.5}$Ti$_3$Zr$_{0.5}$NbMo$_{0.2}$ is 57.7%, and the molar content

 

of Al is 9.6%. After 40 min of constant potential polarization in the primary passivation zone (1 $V_{SCE}$), the molar contents of Ti and Al elements on the sample surface ($d = 0$ nm) were 61.82% and 12.96%, respectively (Table S4), and the molar contents of Ti and Al elements in the interior of the sample close to the passivation layer ($d = 20$ nm) were 57% and 12%, respectively. After 40 min of constant potential polarization in the secondary passivation zone (4$V_{SCE}$), the molar contents of Ti and Al on the surface of the samples ($d = 0$ nm) were 68.82% and 11.29%, respectively, and the molar contents of Ti and Al elements inside the samples close to the passivation layer ($d = 20$ nm) were 62% and 13.37%, respectively. From the moment of corrosion initiation to the moment of primary passivation of the Tafel curve and then to the moment of secondary passivation, the Ti and Al molar content on the surface of the sample gradually increases, as well as those inside the sample near the passivation film. This indicates that outward diffusion of Ti and Al inside the sample occurred and was upslope. The molar content of Zr in the high-entropy alloy samples before and after corrosion was about 10% (Table S4), so no uphill diffusion of Zr occurred. From the morphology of the passivation film in Fig. 8, it can be seen that the distribution of the passivation film is uniform. According to the surface morphology after dynamic potential polarization, it can be introduced that after reaching the breakdown potential, the passivation film at the grain boundary will be dissolved first due to the high energy. After reaching a certain concentration, Ti/Al atoms diffuse uphill and O continues to diffuse downhill, which leads to the formation of a layered structure of the passivation film. Furthermore, the oxides formed by the elements Nb and Ti are typically amorphous or microcrystalline[41,61], which restricts the channels required for diffusion, effectively blocking the entry of aggressive ions and oxygen. This also serves to slow down and prevent metal dissolution. The oxyphilicity of a metallic element can be expressed by the electronegativity difference of each metallic element with oxygen. If the electronegativity of oxygen is 3.44, the difference in electronegativity between Al, Ti, Zr, Nb, and Mo and oxygen is 1.83, 1.9, 2.11, 1.84, and 1.28, respectively[65]. The greater the electronegativity difference, the more oxygen-friendly the metallic element is. Mo is the least oxyphilic and has the most positive Gibbs free energy to react with $H_2O$, so Mo is the least likely to react during corrosion. Zr, although having strong oxygeno-philicity and more negative Gibbs free energy, is not easily diffused outward in the corrosion reaction because it has the largest atomic radius. Al, Ti, and Nb all have both strong oxygenophilicity and more negative Gibbs free energies, and thus are more likely to react with O adsorbed on the sample surface. The Al and Ti atoms have relatively small radii and are more likely to diffuse to the surface to form a passivation film. the Nb atoms have a relatively large radius, which promotes outward diffusion of small radius atoms.

The mechanism of synergistic passivation of $Al_{0.5}Ti_3Zr_{0.5}NbMo_{0.2}$ high entropy alloys Ti and Al in 3.5% NaCl solution can be described as follows. Initially, the adsorption of $H_2O$ under the action of potential generates $TiO_2$ and $Al_2O_3$ on the surface. Due to the insufficiency of oxidant, TiO is generated internally. The crystal structure of $TiO_2$ generated on the surface of the passivation film is composed of octahedra connected with Ti in the center. The voids between the Ti-O octahedra form a channel conducive to the diffusion of oxygen and $Cl^-$[61]. This results in the diffusion of O into the matrix. As corrosion progresses, the oxidant penetrates the interior of the passivation film via the octahedral gaps of $TiO_2$ to engage in an oxidative reaction with $TiO_2$ to form $Ti_2O_3$ and $TiO_2$. As the potential increases, some of the dissolved oxides in the transition zone are oxidized by $Cl^-$ to form $Ti_2O_3$ and $TiO_2$, resulting in a sudden increase in current. Subsequently, the secondary passivation process involves the introduction of additional oxygen, which facilitates the reaction between Ti and oxygen, resulting in the formation of $TiO_2$. The inner layer of $Ti_2O_3$ and TiO undergoes gradual oxidation into $TiO_2$, leading to an increase in the thickness of the passivation film. This, in turn, enhances the uniformity of the distribution of $TiO_2$ in the outermost passivation film.

In summary, $Al_{0.5}Ti_3Zr_{0.5}Nb_xMo_{0.2}$ high entropy alloys are prepared by vacuum arc melting technology, and the corrosion behavior of $Al_{0.5}Ti_3Zr_{0.5}NbMo_{0.2}$ high entropy alloy in 3.5% NaCl solution, as well as the composition and structure of the passivation film are studied, and some of the conclusions can be summarized as follows: 1. The $Al_{0.5}Ti_3Zr_{0.5}Nb_xMo_{0.2}$ high-entropy alloy, with a single-phase BCC structure, exhibits superior corrosion resistance, a wide and stable passivation zone, and breakdown potentials of up to 8.68 $V_{SCE}$. 2. $Al_{0.5}Ti_3Zr_{0.5}NbMo_{0.2}$ high-entropy alloys exhibit preferential corrosion at the {011} grain surface due to the optimal grain orientation. An increase in the molar content of Nb decreases the degree of lattice distortion and decreases the intergranular spacing of the {011} crystal plane spacing, slowing down the corrosion process. This leads to an increase in the alloy's self-corrosion potential, a reduction in its self-corrosion current density, and an improvement in corrosion resistance. 3. Nb, due to its large atomic radius and strong affinity for oxygen, prompts uphill diffusion of Ti/Al atoms of small radius to form stable passivation films such as $Al_2O_3$/$TiO_2$ on the surface. The passivation films such as TiO, NbO in the inner layer formed firstly continuously react with inwardly diffusing O to form highly stable $TiO_2$, $Al_2O_3$ passivation film and a small amount of $Nb_2O_5$.

## Methods

### Materials preparation

$Al_{0.5}Ti_3Zr_{0.5}Nb_xMo_{0.2}$ high entropy alloy produced by non-self-consumption vacuum arc melting furnace. Dynamic potential polarization tests were conducted from −1 to 10 $V_{SCE}$ at a scan rate of 1 mV/s using a 1 $cm^2$ contact area in a 3.5% NaCl solution. The constant potential tests were carried out at 1 $V_{SCE}$ and 4 $V_{SCE}$ for 40 min respectively. Each of the tests was repeated three times to make the data reproducible. Passivation film cross-section samples were prepared using the Focus on Ion Beam Thinning (FIB) thinning technique. The steps of transmission sample preparation included: 1. The discs were cut from the plate-like high entropy alloy using a wire cutter, with a thickness of 3 mm and a width of 0.5 mm. They were then polished to a thickness of 80 μm. 2. The discs were placed into a 3 The discs were immersed in a 3.5% NaCl solution under constant potential polarization at 4 $V_{SCE}$ for 40 min. Subsequently, the discs were thinned on one side (the film-free side) using an ion thinning apparatus.

### Phase and microstructure characterization

Crystal phase identification was performed using an XRD (D8ADVANCE-A25) with an accelerating voltage of 40 KV, Cu Kα source ($\lambda = 0.1542$ nm), diffraction angle of 20° - 80° and scanning rate of 0.02°/s. Using a field emission scanning electron microscope (FEI Nova Nano SEM450) equipped with an energy dispersive spectrometer (EDS), the high entropy alloys were analyzed. The phase structure and composition of the high-entropy alloys were investigated using transmission electron microscopy (TEM, Talos F200X), selected area electron diffraction (SAED), and energy-dispersive spectrometry (EDS).

### Passivation film characterization

Scanning electron microscopy was employed to examine the surface and cross-sectional morphology of the samples following electrochemical testing. The passivation films resulting from constant potential testing were analyzed using an X-ray photoelectron spectrometer (XPS, Thermo Scientific K-Alpha) with an Al Kα-ray source (hv = 1486.6 eV), a vacuum level of the analysis chamber of better than $2 \times 10^{-7}$ mBar, a spot size of 400 μm, and an operating voltage of The applied voltage was 12 kV, the filament current was 6 mA. The full-spectrum scanning fluence was 150 eV in steps of 1 eV, while the fine-spectrum scanning fluence was 50 eV in steps of 0.1 eV. The surfaces were etched with argon ions at depths of 2.5 nm, 10 nm, and 20 nm, respectively, at an etching rate of 0.29 nm/s (with respect to a standard sample of $Ta_2O_5$), and the surfaces were subsequently subjected to X-ray photoelectron spectroscopy (XPS)

testing at the surfaces as well as at the different sputtering depths. All binding energies were calibrated using C1s, 284.8 eV, and the test results were fitted and analyzed using Advantage software. The structure and composition of the passivation films were observed and analyzed using transmission electron microscopy.

## Data availability

Source data are provided with this paper. The data generated in this study are provided in the Supplementary Information/Source Data file. Source data are provided with this paper.

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

## Acknowledgements

This work was supported by the National Natural Science Foundation of China (52161028) (for Q.C.), Academic and the Technical Leaders Program of Major Disciplines of Jiangxi Province (20213BCJ22017) (for Q.C.).

## Author contributions

X.Y. conceived the study and completed the manuscript writing of the paper. Q.C. supervised the project and analyzed the data. X.C. provided the imaging technique support on the SEM and TEM. D.O. analyzed the data and co-wrote the manuscript.

## Competing interests

The authors declare no competing interests.
