## [Transparent Peer Review file · Nature Communications]

Nb-induced lattice changes to enhance corrosion resistance of $\text{Al}_{0.5}\text{Ti}_3\text{Zr}_{0.5}\text{Nb}_x\text{Mo}_{0.2}$ high-entropy alloys

Corresponding Author: Professor Qingjun Chen

Version 0:

Reviewer comments:

Reviewer #1

(Remarks to the Author)

The paper is entitled "Nb-induced lattice changes to enhance corrosion resistance of $\text{Al}_{0.5}\text{Ti}_3\text{Zr}_{0.5}\text{Nb}_x\text{Mo}_{0.2}$ high-entropy alloys, however, the lattice distortion was not calculated with changed Nb, e.g., the atomic size difference, please consult : Xuehui Yan et al 2022 Mater. Futures 1 022002

Reviewer #2

(Remarks to the Author)

This manuscript reports the preparation of a lightweight and highly corrosion-resistant single-phase BCC $\text{Al}_{0.5}\text{Ti}_3\text{Zr}_{0.5}\text{Nb}_x\text{Mo}_{0.2}$ ($x=0.5,0.8,1$) high entropy alloy using vacuum arc melting method. The alloy preferentially corrodes on the {011} crystal plane, and the increase in Nb content reduces the spacing between {011} crystal planes. Therefore, the $\text{Al}_{0.5}\text{Ti}_3\text{Zr}_{0.5}\text{Nb}_x\text{Mo}_{0.2}$ high entropy alloy exhibits the best corrosion resistance in a 3.5% NaCl solution. The XPS and TEM results indicate that the high entropy alloy $\text{Al}_{0.5}\text{Ti}_3\text{Zr}_{0.5}\text{Nb}_x\text{Mo}_{0.2}$ forms a stable passivation film. The corrosion rate of the secondary passivation film is lower than that of the primary passivation film, and the passivation film has a layered structure. The surface of the passivation film is mainly composed of high valence nano oxides such as TiO_2 , Nb_2O_5 , ZrO_2 , etc. From the surface to the interior, the high valence oxides gradually decrease, the low valence oxides gradually increase, and the amorphous oxides increase. The increase in Nb content increases the content of Nb_2O_5 in the passivation film, improving the corrosion resistance of high entropy alloys. Thermodynamic calculations show that the passivation film competes for growth, and the passivation film has high entropy properties, resulting in the formation of an amorphous passivation film that effectively blocks the invasion of ions, thereby improving the corrosion resistance of the material. This work provides excellent data support and theoretical guidance for understanding the internal laws between atomic composition, organizational state, and corrosion resistance mechanism of high entropy alloys. This manuscript is well-written. I therefore strongly recommend it to be accepted for publication in Nature Communications. The following minor points are suggested to be considered:

1 The Conclusion and Abstract section is too data-driven and abstract, which is not conducive to reading. It is recommended that the author condense and summarize the corrosion mechanism in the Conclusion and Abstract section.

2 All abbreviations in the manuscript should be defined at their first mention, such as TEM, FFT, XPS, SEM.

3. All variables, such as I_{corr} and E_{corr} should be defined at their first mention.

Reviewer #3

(Remarks to the Author)

This paper focuses on tailoring the composition of a lightweight refractory high entropy alloy to enhance its corrosion resistance. The method introduced is by adding different amount of Nb to the alloy system and the fundamental mechanism lies in manipulating the crystalline plane spacing as well as the oxide formation process. Although the results can provide new information to the relevant research field in terms of designing high entropy alloys with improved corrosion properties, the novelty of the work is not high and the advancement in methods or knowledge is also limited. A few other critical points

are listed below.

1. The authors reported that the designed alloys exhibit a low passivation current density as low as $7.07 \times 10^{-6} \text{A/cm}$. However, this current density seems to be within the normal range for metals and alloys with decent corrosion resistance property. For example, the table below shows the passivation current density for some widely researched metals and alloys. It would be good to include a comparison between this work and the previously published work.

Metal/Alloy Passivation Current Density (A/cm^2) Notes

Titanium (Ti) 1×10^{-8} - 1×10^{-7} Forms stable oxide layer, excellent for biomedical applications

Titanium Alloys (e.g., Ti-6Al-4V) 1×10^{-7} - 5×10^{-7} Common in medical implants, aerospace

Stainless Steel (316L) 1×10^{-7} - 1×10^{-6} Widely used in marine, biomedical, and food industries

Stainless Steel (304) 5×10^{-7} - 5×10^{-6} Less corrosion-resistant than 316L

Nickel (Ni) 5×10^{-7} - 1×10^{-5} Forms passive oxide layer, useful in high-temperature applications

Nickel Alloys (e.g., Inconel 625) 1×10^{-8} - 1×10^{-6} High resistance to oxidation and corrosion at high temperatures

Chromium (Cr) 1×10^{-7} - 1×10^{-6} Key component in stainless steels

Aluminum (Al) 1×10^{-7} - 5×10^{-6} Forms stable oxide layer, often used with surface treatments

Aluminum Alloys (e.g., 6061) 1×10^{-6} - 1×10^{-5} Common in structural applications

Tantalum (Ta) $< 1 \times 10^{-8}$ Extremely resistant, used in chemical process equipment

Zirconium (Zr) $< 1 \times 10^{-8}$ - 1×10^{-7} High resistance to corrosion, used in nuclear reactors

2. Regarding the mechanisms for enhancing the corrosion performance of the designed alloys, the authors focus on two aspects. One is the change of {011} crystal plane spacing by adding Nb which can reduce the grain surface spacing and hence hindering corrosion. However, the authors did not give detailed explanation why adding Nb can decrease the crystal plane spacing {011}.

3. If it is related to the radius of the element, then Mo and Zr shall also have similar influences due to their radius mismatch. What is the interplay among these elements in terms of plain spacing and corrosion process?

4. In addition, what is the chemical distribution of these elements? If some elements segregate along the grain boundaries or have preferred distribution along certain crystal plains, will the distribution of elements also play a role here? Therefore, more work needs to be completed to elucidate this.

5. Following comment 2 above, the other mechanism proposed in this work is about the oxide formation through promoting the uphill diffusion of Ti/Al and downhill diffusion of O so a layered structure can be formed. However, the authors did not provide enough discussion here to facilitate the understanding of the proposed mechanisms.

6. For instance, why Ti and Al showed an uphill diffusion behaviour? What about Zr? Is this layered structure homogeneous and not influenced by the grain boundaries? The grain boundary shall also promote the diffusion of O or other elements.

Version 1:

Reviewer comments:

Reviewer #1

(Remarks to the Author)

The authors have addressed all the comments raised by the reviewers, it can be accepted now.

Reviewer #2

(Remarks to the Author)

The authors have carefully revised this manuscript. I therefore recommend it to be accepted for publication in Nature Communications.

Reviewer #3

(Remarks to the Author)

Only one minor suggestion is regarding oxyphilicity. A reference should be added to the discussion section regarding oxyphilicity to ensure the original data and scientific findings are accessible.

Reviewer #1:

Dear reviewer:

Thank you very much for your comments and professional advice. These opinions help to improve academic rigor of our article based on your suggestion and request, we have made corrected modifications on the revised manuscript. We hope that our work can be improved again. Furthermore, we would like to show the details as follows:

	Question	Response
Question1	The paper is entitled "Nb-induced lattice changes to enhance corrosion resistance of $\text{Al}_{0.5}\text{Ti}_3\text{Zr}_{0.5}\text{Nb}_x\text{Mo}_{0.2}$ high-entropy alloys, however, the lattice distortion was not calculated with changed Nb, e.g., the atomic size difference, please consult: Xuehui Yan et al 2022 Mater. Futures 1 022002.	The thermodynamic parameters obtained according to the thermodynamic formulae (1)~(6) are listed in Table 1. The atomic radius differences δ_r are 3.06%, 3.11% and 3.2%, respectively. The cell parameters of the high entropy alloy can be obtained from XRD refinement as 3.299Å, 3.294Å, and 3.293Å, respectively. Meanwhile, the TEM in Fig. 2 demonstrates that the lattice spacing of the high-entropy alloy $\text{Al}_{0.5}\text{Ti}_3\text{Zr}_{0.5}\text{Nb}_x\text{Mo}_{0.2}$ (110) is 2.282 Å, 2.289 Å, and 2.295 Å. All of the above results show that the high-entropy alloy produces lattice alteration with the change of Nb content. The relevant text has been amended in the main text, lines 107-124.

Reviewer #2:

Dear reviewer:

Thank you very much for your recognition and professional advice on this work. These opinions help to improve academic rigor of our article based on your suggestion and request, we have made corrected modifications on the revised manuscript. We hope that our work can be improved again. Furthermore, we would like to show the details as follows:

	Question	Response
Question1	The Conclusion and Abstract section is too data-driven and abstract, which is not conducive to reading. It is recommended that the author condense and summarize the corrosion mechanism in the Conclusion and Abstract section.	The summary has been revised: In this work, the effect of lattice structure on the corrosion behavior and passivation film properties of reinforced $\text{Al}_{0.5}\text{Ti}_3\text{Zr}_{0.5}\text{Nb}_x\text{Mo}_{0.2}$ ($x=0.5, 0.8, 1$) high-entropy alloys was investigated. A single-phase BCC $\text{Al}_{0.5}\text{Ti}_3\text{Zr}_{0.5}\text{Nb}_x\text{Mo}_{0.2}$ ($x=0.5, 0.8, 1$) high-entropy alloy, exhibiting remarkable corrosion resistance, was synthesized using vacuum arc melting. Nb improves the corrosion resistance of high-entropy alloys in two main ways. On the one hand, the alloy showed preferential corrosion at the $\{011\}$ crystalline planes. Increasing Nb content reduced the $\{011\}$ crystalline plane spacing, enhancing the corrosion resistance of $\text{Al}_{0.5}\text{Ti}_3\text{Zr}_{0.5}\text{NbMo}_{0.2}$. On the other hand, On the other hand, during the corrosion process, Nb, which has a large atomic radius and strong oxygenophilicity, interacts with each metal element, contributing to the uphill

		diffusion of Al/Ti and the downhill diffusion of O. The low-valent oxides formed first continuously react with the inward-diffusing O to form high-valent oxides. This results in the formation of a layered passivation film with high breakdown potential and high stability. This work provides a basis for designing chemically robust alloys for extreme environments. Corrosion mechanisms have been restated in the abstract and conclusions and are highlighted in red. The relevant text has been amended in the main text, lines 11-20.
		Conclusions (2) and (3) have been modified: (2) High-entropy alloys exhibit preferential corrosion at the {011} grain surface due to the optimal grain orientation. An increase in the molar content of Nb decreases the degree of lattice distortion and decreases the intergranular spacing of the {011} crystal plane spacing, slowing down the corrosion process. This leads to an increase in the alloy's self-corrosion potential, a reduction in its self-corrosion current density, and an improvement in corrosion resistance. (3) Nb, due to its large atomic radius and strong affinity for oxygen, prompts uphill diffusion of Ti/Al atoms of small

		radius to form stable passivation films such as $\text{Al}_2\text{O}_3/\text{TiO}_2$ on the surface. The passivation films such as TiO, NbO in the inner layer formed firstly continuously react with inwardly diffusing O to form highly stable TiO_2, Al_2O_3 passivation film and a small amount of Nb_2O_5. The relevant text has been amended in the main text, lines 468-470,473-477.
Question2	All abbreviations in the manuscript should be defined at their first mention, such as TEM, FFT, XPS, SEM.	All abbreviations in the manuscript have been defined in the text. The relevant text has been amended in the main text, lines 97-98, 101-102, 142, 144, 485, 505.
Question3	All variables, such as I_{corr} and E_{corr} should be defined at their first mention.	All variables in the manuscript have been defined in the text. The relevant text has been amended in the main text, lines 43-44.

Reviewer #3:

Dear reviewer:

Thank you very much for your comments and professional advice. These opinions help to improve academic rigor of our article based on your suggestion and request, we have made corrected modifications on the revised manuscript. We hope that our work can be improved again. Furthermore, we would like to show the details as follows:

	Question	Response
Question1	The authors reported that the designed alloys exhibit a low passivation current density as low as 7.07×10^{-6} A/cm. However, this current density seems to be within the normal range for metals and alloys with decent corrosion resistance property. For example, the table below shows the passivation current density for some widely researched metals and alloys. It would be good to include a comparison between this work and the previously published work. Metal/Alloy Passivation Current Density (A/cm²) Notes Titanium (Ti) 1×10^{-8} - 1×10^{-7} Forms stable oxide layer, excellent for biomedical applications	The $\text{Al}_{0.5}\text{Ti}_3\text{Zr}_{0.5}\text{NbMo}_{0.2}$ high entropy alloy designed in this work is notably characterized by a very high breakdown potential, along with a low self-corrosion current density and a dimensional passivation current density comparable to that of conventional corrosion-resistant alloy materials. The passivation film is more stable than other conventional alloys as well as high entropy alloys of other systems. This results in excellent overall corrosion resistance. Comparison of $E_{\text{pit}}-I_{\text{corr}}$ plots of corrosion of $\text{Al}_{0.5}\text{Ti}_3\text{Zr}_{0.5}\text{NbMo}_{0.2}$ high-entropy alloy with titanium alloys, nickel-based alloys, and high-entropy alloys such as $\text{Al}_x\text{CoCrFeNi}$ system and NbMoZrTiAlV_x system corroded under 3.5% NaCl solution are listed in Fig. 4b. The relevant text has been amended in the main text, lines 164-167, 704-705.

Titanium Alloys (e.g., Ti-6Al-4V) 1×10^{-7} - 5×10^{-7}
 Common in medical implants, aerospace
 Stainless Steel (316L) 1×10^{-7} - 1×10^{-6} Widely used in marine, biomedical, and food industries
 Stainless Steel (304) 5×10^{-7} - 5×10^{-6} Less corrosion-resistant than 316L
 Nickel (Ni) 5×10^{-7} - 1×10^{-5} Forms passive oxide layer, useful in high-temperature applications
 Nickel Alloys (e.g., Inconel 625) 1×10^{-8} - 1×10^{-6} High resistance to oxidation and corrosion at high temperatures
 Chromium (Cr) 1×10^{-7} - 1×10^{-6} Key component in stainless steels
 Aluminum (Al) 1×10^{-7} - 5×10^{-6} Forms stable oxide layer, often used with surface treatments
 Aluminum Alloys (e.g., 6061) 1×10^{-6} - 1×10^{-5}
 Common in structural applications
 Tantalum (Ta) $< 1 \times 10^{-8}$
 Extremely resistant, used in

	chemical process equipment Zirconium (Zr) $< 1 \times 10^{-8} - 1 \times 10^{-7}$ High resistance to corrosion, used in nuclear reactors	
Question2	Regarding the mechanisms for enhancing the corrosion performance of the designed alloys, the authors focus on two aspects. One is the change of {011} crystal plane spacing by adding Nb which can reduce the grain surface spacing and hence hindering corrosion. However, the authors did not give detailed explanation why adding Nb can decrease the crystal plane spacing {011}.	The atomic radii of five elements, Al, Ti, Zr, Nb and Mo, are in the order of $r(\text{Al}) < r(\text{Ti}) < r(\text{Mo}) < r(\text{Nb}) < r(\text{Zr})$, and the thermodynamic parameters of the five elements solidly dissolved in a certain ratio, such as ΔS_{mix} (entropy of mixing), ΔH_{mix} (enthalpy of mixing), VEC (valence electron concentration), and δ_r (difference in atomic radius), were calculated and supplemented in Table 1. where the atomic radius difference δ_r is 3.2%, 3.11%, and 3.06% when $x=0.5, 0.8, 1$, respectively. This indicates that increasing the molar content of Nb decreases the atomic radius difference of the high-entropy alloy, reducing the extent of lattice distortion and therefore changing the crystal plane spacing of (110). The relevant text has been amended in the main text, lines 327-336.
Question3	If it is related to the radius of the element, then Mo and Zr shall also have similar influences due to their radius mismatch. What is the	The change in crystal spacing is due to differences in the atomic radii of the elements (see Table 1, Table 2). The atomic radii and electronegativity of the five metallic elements are listed in the

	interplay among these elements in terms of plain spacing and corrosion process?	table below. The oxyphilicity of a metallic element can be expressed by the electronegativity difference of each metallic element with oxygen. If the electronegativity of oxygen is 3.44, the difference in electronegativity between Al, Ti, Zr, Nb, and Mo and oxygen is 1.83, 1.9, 2.11, 1.84, and 1.28, respectively. The greater the electronegativity difference, the more oxygen-friendly the metallic element is. Mo is the least oxyphilic and has the most positive Gibbs free energy to react with H₂O, so Mo is the least likely to react during corrosion. Zr, although having strong oxygenophilicity and more negative Gibbs free energy, is not easily diffused outward in the corrosion reaction because it has the largest atomic radius. Al, Ti, and Nb all have both strong oxygenophilicity and more negative Gibbs free energies, and thus are more likely to react with O adsorbed on the sample surface. The Al and Ti atoms have relatively small radii and are more likely to diffuse to the surface to form a passivation film. the Nb atoms have a relatively large radius, which promotes outward diffusion of small radius atoms. The relevant text has been amended in the main text, lines 428-
--	--	---

		441.
Question4	In addition, what is the chemical distribution of these elements? If some elements segregate along the grain boundaries or have preferred distribution along certain crystal plains, will the distribution of elements also play a role here? Therefore, more work needs to be completed to elucidate this.	We have added data on the microstructure and morphology of the as-cast high-entropy alloys in the paper. From the SEM-EDS mapping of the high-entropy alloy (Fig. 3), it can be seen that the chemical distribution of the five elements is uniform and does not aggregate along the grain boundaries. Therefore, the elemental chemical distribution does not play an influential role in the corrosion process. The relevant text has been amended in the main text, lines 137-140.
Question5-6	Following comment 2 above, the other mechanism proposed in this work is about the oxide formation through promoting the uphill diffusion of Ti/Al and downhill diffusion of O so a layered structure can be formed. However, the authors did not provide enough discussion here to facilitate the understanding of the proposed mechanisms. For instance, why Ti and Al showed an uphill diffusion behaviour? What about Zr? Is this layered structure homogeneous and not	According to the above SEM-EDS experimental results, it can be seen that the chemical distribution of the elements at each place of the high-entropy alloy is uniform, so at the initial moment of corrosion, the molar content of Ti on the surface and in the interior of the high-entropy alloy $\text{Al}_{0.5}\text{Ti}_3\text{Zr}_{0.5}\text{NbMo}_{0.2}$ is 57.7%, and the molar content of Al is 9.6%. After 40 min of constant potential polarization in the primary passivation zone (1 V_{SCE}), the molar contents of Ti and Al elements on the sample surface (d=0 nm) were 61.82% and 12.96%, respectively (Table 2), and the molar contents of Ti and Al elements in the interior of the sample close to the passivation layer (d=20 nm) were 57%

	influenced by the grain boundaries? The grain boundary shall also promote the diffusion of O or other elements.	and 12%, respectively. After 40 min of constant potential polarization in the secondary passivation zone ($4V_{SCE}$), the molar contents of Ti and Al on the surface of the samples ($d=0nm$) were 68.82% and 11.29%, respectively, and the molar contents of Ti and Al elements inside the samples close to the passivation layer ($d=20nm$) were 62% and 13.37%, respectively. From the moment of corrosion initiation to the moment of primary passivation of the Tafel curve and then to the moment of secondary passivation, the Ti and Al molar content on the surface of the sample gradually increases, as well as those inside the sample near the passivation film. This indicates that outward diffusion of Ti and Al inside the sample occurred and was upslope. The molar content of Zr in the high-entropy alloy samples before and after corrosion was about 10% (Table 2), so no uphill diffusion of Zr occurred. From the morphology of the passivation film in Fig. 7, it can be seen that the distribution of the passivation film is uniform. According to the surface morphology after dynamic potential polarization (Fig. 3), it can be introduced that after reaching the breakdown potential, the passivation film at the grain boundary will be
--	--	--

		dissolved first due to the high energy. The relevant text has been amended in the main text, lines 400-423.
--	--	--

Reviewer #3:

Dear reviewer:

Thank you very much for your comments and professional advice. These opinions help to improve academic rigor of our article based on your suggestion and request, we have made corrected modifications on the revised manuscript. We hope that our work can be improved again. Furthermore, we would like to show the details as follows:

	Question	Response
Question1	Only one minor suggestion is regarding oxyphilicity. A reference should be added to the discussion section regarding oxyphilicity to ensure the original data and scientific findings are accessible.	A reference to oxyphilicity has been inserted on line 434. Qteish A. Electronegativity scales and electronegativity-bond ionicity relations: A comparative study. J. Phys. Chem. Solids 124, 186-191 (2019). The oxyphilicity of a metal element is expressed as the difference in electronegativity between the metal element and the O element. Specific values for the electronegativity of the O element and the metal element are available in the literature. This ensures that the original data and scientific findings are accessible.